# A Trustworthy Robot Buddy for Primary School Children

**Matthijs H. J. Smakman** [1,2,*], **Daniel F. Preciado Vanegas** [2], **Koen Smit** [1], **Sam Leewis** [1], **Youri Okkerse** [1], **Jesper Obbes** [1], **Thom Uffing** [1], **Marina Soliman** [1], **Tony van der Krogt** [1] and **Lucas Tönjes** [1]

1   Institute for Information Communication and Technology, HU University of Applied Sciences Utrecht, Heidelberglaan 15, 3584 CS Utrecht, The Netherlands; koen.smit@hu.nl (K.S.); sam.leewis@hu.nl (S.L.); youri.okkerse@student.hu.nl (Y.O.); jesper.obbes@student.hu.nl (J.O.); thom.uffing@student.hu.nl (T.U.); marina.soliman@student.hu.nl (M.S.); tony.vanderkrogt@student.hu.nl (T.v.d.K.); lucas.tonjes@student.hu.nl (L.T.)
2   Department of Communication Science, VU University Amsterdam, De Boelelaan 1105, 1081 HV Amsterdam, The Netherlands; d.f.preciadovanegas@vu.nl
*   Correspondence: matthijs.smakman@hu.nl or m.h.j.smakman@vu.nl

**Abstract:** Social robots hold potential for supporting children's well-being in classrooms. However, it is unclear which robot features add to a trustworthy relationship between a child and a robot and whether social robots are just as able to reduce stress as traditional interventions, such as listening to classical music. We set up two experiments wherein children interacted with a robot in a real-life school environment. Our main results show that regardless of the robotic features tested (intonation, male/female voice, and humor) most children tend to trust a robot during their first interaction. Adding humor to the robots' dialogue seems to have a negative impact on children's trust, especially for girls and children without prior experience with robots. In comparing a classical music session with a social robot interaction, we found no significant differences. Both interventions were able to lower the stress levels of children, however, not significantly. Our results show the potential for robots to build trustworthy interactions with children and to lower children's stress levels. Considering these results, we believe that social robots provide a new tool for children to make their feelings explicit, thereby enabling children to share negative experiences (such as bullying) which would otherwise stay unnoticed.

**Keywords:** social robot; buddy; children; HRI; well-being; trust; education

## 1. Introduction

Social robots are being increasingly introduced in primary education as a tool for teaching children topics such as a second language [1,2], mathematics [3], and geography [4]. Although social robots hold potential as tutors, there are multiple technological and ethical challenges for the large-scale introduction of social robots as effective, capable teaching tools for primary education. Short-term, the benefits of these robots may lay more in increasing children's well-being, than in their ability to increase learning gains. However, while it is still hard to evaluate robots' abilities to directly affect learning outcomes, they might still be able to do so indirectly by increasing well-being and motivational factors in students.

Two important societal issues for primary school children's well-being are bullying and stress. A recent study conducted by five universities in the Netherlands concluded that almost one out of three children is bullied during their primary education [5]. A substantial amount of these children that are being bullied (40%) do not share this fact with an adult [6]. Consequently, such problems often stay unnoticed. The impacts of being bullied are not to be underestimated, because as a result children could feel a significant increase of loneliness, lack of confidence, and even feel depressed [7]. This may in turn lead to increased levels of stress, which can have a negative impact on several physiological and

psychological functions, especially in children who are still in their developmental stage [8]. The effects of high levels of stress are not limited to childhood, because stress is considered an important factor in health problems, violence, and academic failure in adulthood [9]. Therefore, bullying and stress in primary schools are two important societal issues that require both public and academic attention.

Social robots can have multiple benefits related to children's well-being, both in healthcare settings [10,11], as well as in therapy and education [12]. Social robots can be used to help children talk about problems and experiences they do not feel comfortable sharing with other people. Children are also reported to be willing to share secrets with social robots [13]. Children, in general, are more likely to share their secrets with someone they trust, according to [14,15].

Children with Autism Spectrum Disorder (ASD) are one of the major target groups in the Human–Robot Interaction literature [16–18]. Children with ASD often have a difficult time communicating with other children. Studies have shown that children with ASD are more willing to talk to a social robot than to other people [19,20]. Having children share their social issues with robots can potentially assist in finding the necessary help they need to overcome the struggles that they are experiencing.

The aim of this study was twofold; (1) to explore if a social robot is able to reduce children's stress levels in a real-life primary school environment, and (2) to examine the effect of intonation, gender, and humor—elements that play an important role in gaining someone's trust in a social robot [1,21–23]. In the following section, these trust elements are discussed in further detail. After that, the research method is explained, and the data collection and analysis are presented. This is followed by the results of our study. Lastly, the paper ends with discussion of the main results, limitations, conclusions, and future research directions.

## 2. Background

Social robots in education can take on several roles, such as a tutor [24], peer [25], or buddy [26]. Some of these roles, such as that of a buddy, allow a child to disclose information more easily to a robot than to a human teacher. In earlier research European teachers indicated that children might feel more comfortable expressing their uncertainties to a robot tutor than to a human, due to the lack of fear of judgment [27]. In addition, children have been shown to be significantly more likely to report that classmates were bullied to a social robot interviewer in comparison to the human interviewer [28].

In the Netherlands, 3.9% of primary school children, aged between four and twelve years, are diagnosed with ASD [29]. For these children, relevant findings have been presented in the past. For example, the study from Dautenhan et al. [30] shows how social robots can be used as a social mediator in therapy for children diagnosed with ASD. They describe how social robots can be used in the context of communication, and in this case, as a social mediator between children. Besides the application of social robots in the context of communication, Ref. [30] also showed that the social robot could have a secondary function, such as being used as a playmate by children.

Although the current literature suggests that social robots can be considered useful tools for letting children express their feelings, thereby potentially allowing them to discuss that they are victims of bullying behavior, there are also concerns. Special education teachers, for example, have been reported to view social robots as a potential target for being bullied by children, or even the robots becoming bullies themselves. Children bullying or abusing robots has indeed been reported in HRI studies, such as [31].

For children to open up to social robots they need to believe a robot is believable and trustworthy. For this study we take a broad notion of trust, that is, trust as a belief that the social robot is sincere and will keep its word or promises. This definition largely fits within the definition of social and competency trust as defined by [32], and is in line with other concepts of trust used in HRI [33]. Although the influence of specific robot (or

human) capabilities on trust is difficult to measure, there have been studies that found interesting results.

Earlier research has shown that there is a positive relationship between trust and humor, and for humor to be effective, facial expression and the intonation of a person's voice should first be able to express emotions correctly [34]. Stock [35] also discusses the relationship between humor and trust, stating that humor plays an important role in trusting one another. Furthermore, Stock [35] included intonation as a factor that plays an important part in trust. Intonation aspects, such as loudness, pitch, and vibration of the vocal cords, all play a part in the transmission of emotion in the words that are being spoken. Whether this is also equally important for child–robot interaction is hard to evaluate based on the current body of knowledge, which is often based on short term interactions with a limited number of participants.

Stereotypical behavior and gender have also been shown to influence trust levels and sympathetic behavior in HRI research. For example, ref. [36] showed that a social robot with a male voice was trusted more when the robot had to perform a stereotypical male task for the human, with the female robot being trusted more when it was performing a stereotypical female task. Results of other research also found gender to be of influence on trust levels in HRI; Siegel et al. [22] showed that a social robot from the opposite gender to the user to be more credible, trustworthy, and interesting.

The embodiment of the social robot has also been shown to impact trust. A recent meta-analysis investigating the factors influencing the development of trust towards robots in children [32] showed an overall negative effect of human-like attributes on trust, which could be explained by the increased expectancies of children of a human-like robot [32]. Importantly, the study distinguishes between *competency trust,* understood as the extent to which the child can rely on the robot to be able to perform its tasks; and *social trust*, described as the extent to which the child expects the robot to keep its word and fulfil its promises. Interestingly, the described negative effects of human-like features on trust were observed only for *competency*, but not for *social trust*, suggesting that child expectations regarding the robot's performance may influence their trust depending on the specific function or role of the robot. In this sense, human-like features might negatively influence trust in robots fulfilling a clearly defined functional role (e.g., tutors) rather than a social one (e.g., peers). The length of the interaction also impacted trust; Stower et al. (2021) concluded that shorter interaction may lead to higher trust levels. Another factor that greatly affects interaction, collaboration, and acceptance of social robots is the notion of *comfort*, understood as the extent to which these robots inspire a sense of ease, safety, and security in the users [37–39]. Importantly, comfort is associated with trust, in the sense that people are reported to be more willing to trust and collaborate with robots if the interaction is stress-free, and people feel safe and at ease in the presence of the robot [37]. Stress is of great influence on children's well-being. It is also a subject of interest in HRI research [40–42]. The term stress can be differentiated into three types: positive, tolerable, and toxic [43]. Positive stress is the most common. It is exciting in small doses and arises when exciting actions are being performed for the first time, such as on the first day of school. Tolerable stress is less common and occurs in people who are generally nervous by nature. An example of this is an appointment at the hospital. The worst type of stress is toxic stress. Many cases of toxic stress have their origin in traumatic childhood experiences, such as physical or emotional abuse, which become internalized, remain painful, and are constantly present.

One of the traditional ways to cope with stress is listening to classical music. Earlier research has shown that listening to classical music reduces cortisol and lowers blood pressure in the body while releasing dopamine [44,45].

Next to music theory being a tool for reducing stress, multiple HRI studies have shown that social robots are able to lower stress levels of children in medical settings (tolerable stress) [46–48]. In these studies, robots are applied in the role of a buddy or as a distraction

method. Although the capabilities of the robots in these studies are relatively simple, they are already able to reduce tolerable stress levels in children.

Overall, there seem to be several factors contributing to the trustworthiness of social robots and stress reduction. However, there is a clear need to further examine these factors in relation to child–robot interaction, especially in real-life educational settings. This study aimed to explore whether, humor, intonation, and gender play a role in gaining trust when applied to social robots when interacting with children, and if social robots can reduce children's stress levels in primary education. To do so, we set up two experiments, which are discussed in the next section.

**3. Methods**

For this exploratory case study, we created two unique experiments. The first experiment was aimed at exploring children's trust in social robots (trust experiment). The second experiment focused on reducing children's stress levels using a social robot (stress-reducing experiment).

*3.1. Participants*

In total, 115 unique children participated in both experiments combined. In the trust experiment a total of 55 children participated, aged between 4 and 6 years (Mean = 4.56, Median = 5, SD = 0.57), of which 26 were boys and 29 girls.

In the stress-reducing experiment: 60 children participated aged between 3 and 6 years (Mean = 5.06, Median = 5), of which 30 were boys and 30 girls. In this experiment, we compared the stress levels of children who interacted with a social robot to the stress levels of children who received a traditional music theory intervention. To be able to compare the two interventions, 30 children were placed in the robot group and 30 children received the music theory intervention.

No personal data were gathered from our participants other than gender and age. Parents of the children provided consent for their children to participate in this study. All children provided verbal consent and could stop the experiment at any time. The children were recruited from three primary schools in the Netherlands.

*3.2. Materials*

For the trust experiment, two existing questionnaires used in HRI research that included trust constructs [49,50] were combined into a single questionnaire [51]. These questions were then translated to Dutch and, with the help of two primary school teachers, revised so that the questions would be understandable for young children.

The stress reduction experiment used two measures; (1) observation scheme and (2) a 5-point smiley-based Likert Scale (shown in Figure 1), representing strongly agree to strongly disagree. The scale was guided by a smileyometer to help interpret the question in an understandable way for the children [52]. This tool is considered to be one of the fundamental tools for educational and clinical research [53,54]. This process consists of making a statement and then asking participants questions to show their level of agreement with that statement. An observation scheme was used to observe both the robot (experimental) and the music group (control). The observation scheme was used to register emotional signals, such as: smiling, being at ease, showing shyness and timidity, frowning, or seeing to be uncomfortable. Regarding the robot group, a total of five questions related to the use of the robot were asked after the interaction, these five questions were used in earlier studies [49] and have been adopted for this study.

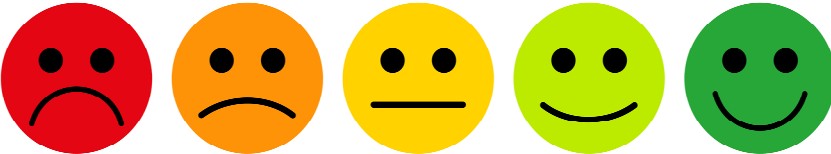

**Figure 1.** Smiley-Based Likert Scale.

### 3.3. Robot

For both experiments, the SAMBuddy Storytelling Cuddle [55] (shown in Figure 2) was utilized. The SAMBuddy Storytelling Cuddle is a stuffed plush animal lookalike robot, filled with very basic hardware components. The main components are a Raspberry Pi Zero, a microphone, and a speaker.

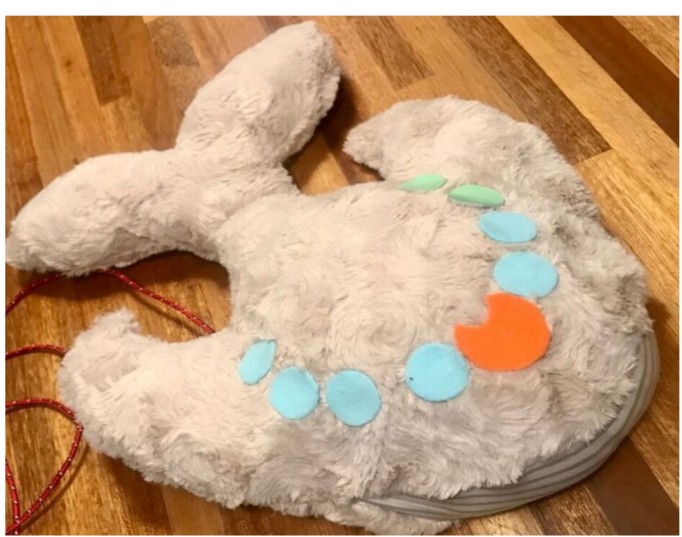

**Figure 2.** SAMBusddy Storytelling Cuddle.

For the trust experiment, the robot was programmed with five different dialogue options: spontaneous male, spontaneous female, monotone male, monotone female, and humorous female. The dialogues were recorded with professional microphones and experienced voice actors. The interaction dialogues consisted of ten lines. The first line included the introduction of the robot and asking the name of the participant. The other nine lines of the dialogue consisted of small talk about the school and what kind of animal the robot was. After every question/response from the robot, the participant had the option to answer. Once having answered, he or she could press a button and the dialogue would continue. The humor variant had extra dialogue. Three child-friendly jokes were added to this version. For privacy reasons, the voice recording option of the robot was disabled. As shown in Figure 2, the robot has several colorful buttons. To give all the children the same interaction the robot was programmed so that regardless of which button the child pressed the robot would proceed with the standard dialogue; the order of the script was fixed.

For the stress reduction experiment, the dialogue entailed a basic introduction to the robot. It then proceeded to ask five questions about the child's likes and hobbies. Among these questions, positive comments or jokes were implemented to relax the child. The robot asked a question and when the child finished answering it, he/she only had to press the orange button to listen to the next question, making it easy to understand for the participants.

### 3.4. Procedure

Before the start of the trust experiment, the research assistant was introduced to the primary class together with the social robot. Once this was done, a child was brought out

of the class to a separate classroom. Here the research assistant conducted a pre-interaction questionnaire regarding sociodemographic data, related to age, grade, gender, if they had ever seen a robot before and if so, their prior experience with them. After this was done, the child started to interact with the robot while the interviewer stayed in the room. Children could obtain a dialogue with a humorous robot, a robot with a monotonous male or female voice, a spontaneous male or spontaneous female voice. The dialogue a child received was selected randomly. Questions of children and assistance were given on the go. Once interaction was over, the questionnaire regarding trust was conducted. Each session lasted on average 10 min.

Regarding the stress experiment, the interviewer was introduced to the class together with the social robot or with an explanation of classical music. Thereafter, a child was brought out of the class to a separate classroom where the child–robot interaction or classical music intervention took place. Here an interviewer conducted a pre-interaction questionnaire regarding sociodemographic data, related to age, grade, and gender. The children in the robot group received some additional questions regarding their prior experience with robots. All participants in this experiment were asked how they currently felt by using the 5-point Smiley-based Likert Scale. For the robot group, this was followed by the child's interaction with the robot, with the interviewer present in the room to assist when needed. For the music theory group, this was followed by one of the interviewers playing the classic song "Comptine d'un autre été (Amélie)". During both interventions (robot and music) a research assistant made observations using an observation scheme. After the music theory intervention or the robot interaction, the same 5-point Smiley-based Likert Scale was used in order to determine the mood and level of stress of the child after the experiment. Each session lasted for approximately 8–10 min. Figure 3 shows a schematic overview of the procedure.

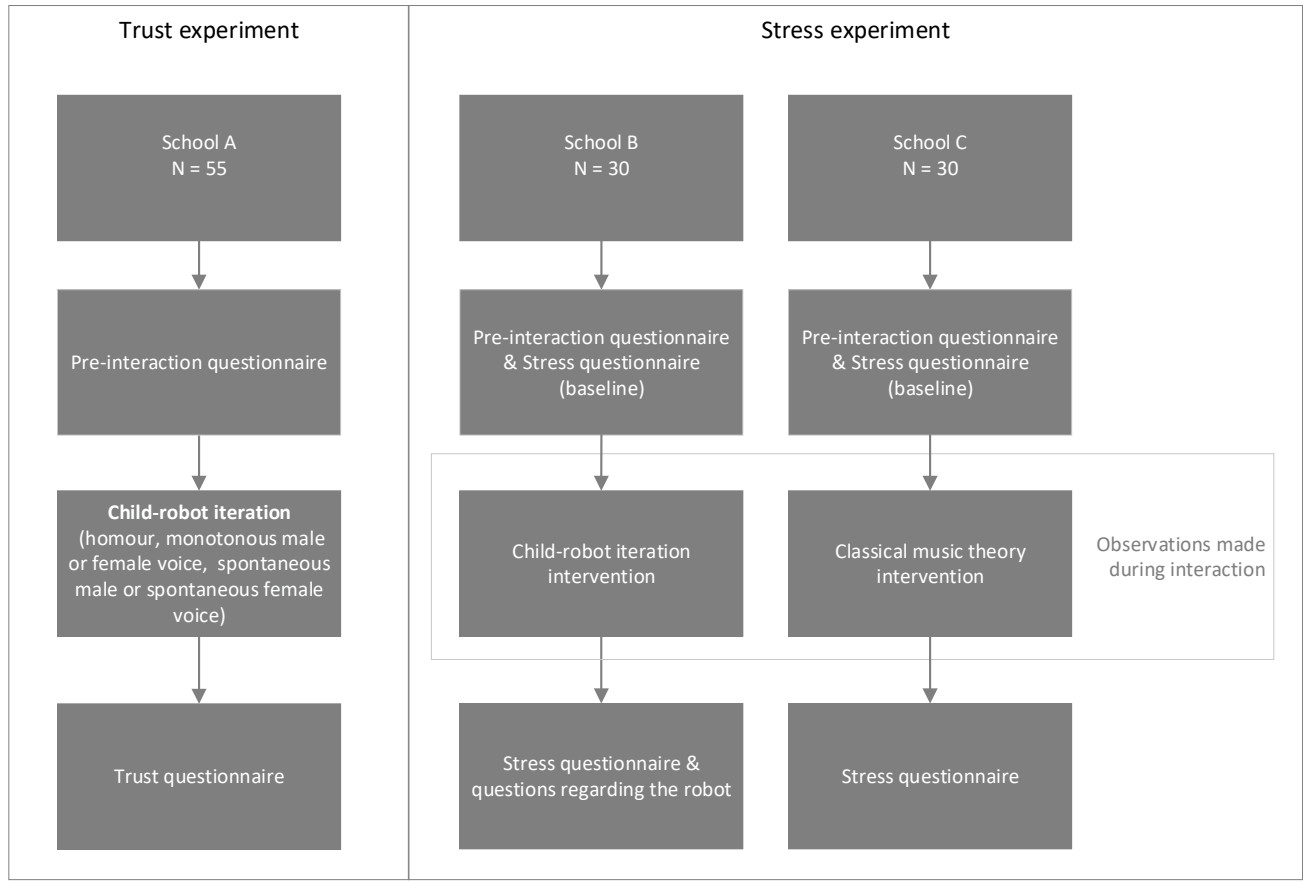

**Figure 3.** Methods of data collection.

## 4. Data Analysis

After the experiments were concluded and the data collection was finalized, the data was transformed for analysis in SPSS (IBM V25).

*The Trust Experiment Scale*

To analyze the results of the trust questionnaire, we first reversed the two items, for which the statements had a negative (rather than a positive) formulation. Second, to check the reliability of our trust scale, we conducted a factor analysis, followed by an internal consistency test (i.e., reliability) of the trust scale using Cronbach's alpha. The initial reliability test resulted in a Cronbach's alpha of $\alpha = 0.448$, which can be considered unreliable [56]. Therefore, Question 3 and Question 6 were removed to raise the reliability of the stress scale to $\alpha = 0.579$; this value can be considered sufficient for exploratory research with our number of items in the trust scale [57]. The final items included in the trust scale can be found online [51]. Finally, a UNIANOVA test was performed comparing gender and robot type to check for differences between trust in robot types with boys or girls.

## 5. Results

In this section, the results of the study are presented in two parts. First, the results of the trust experiment are presented. Second, the results of the stress experiment are presented. The questions, syntax, and dataset are available on the Open Science Framework [51].

*5.1. The Trust Experiment*

To explore children's trust in the social robot we conducted a UNIANOVA test with the trust scale (see data Analysis Section) and factors that were expected to influence trust (robot type, age, grade, teacher, gender, earlier robot experience, and experience with the robot buddy). None of the variables showed a significant $(p < 0.05)$ effect on trust.

Overall, the results showed high levels of trust for all the robot types, shown in Figure 4. Although the differences in trust values between the different robot types were not significant, children trusted the monotonous female voice most, and the robot with humor least.

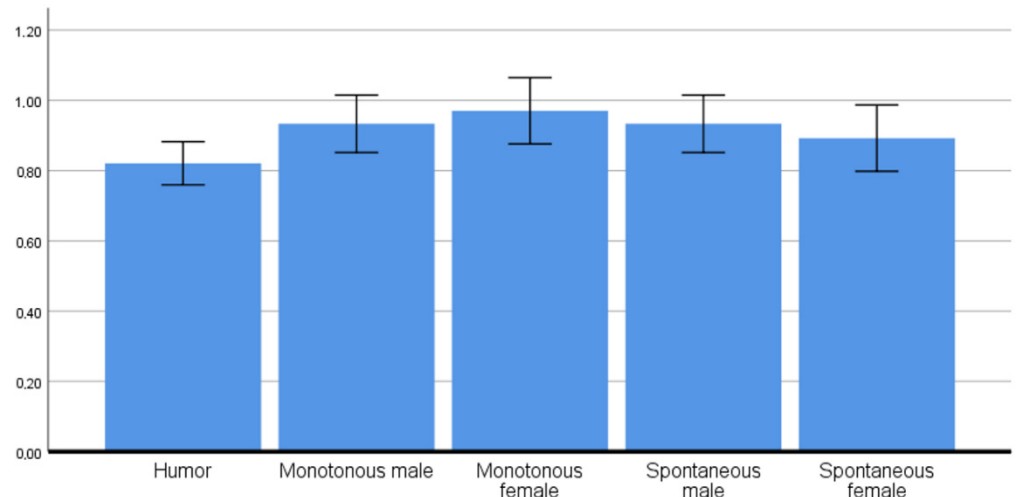

**Figure 4.** Mean scored per type of robot, including error bars.

Although the type of robot did not have a significant effect, it almost showed a trend, $(p \leq 0.1$ is considered a trend), $(p = 0.105)$. Therefore, to explore the effect of gender on stress scores per robot type in more detail, we ran a UNIANOVA test. The results showed a trend $(p = 0.08)$, wherein the trust levels of boys and girls differed for the humorous robot.

Considering the means per type of robot per gender (shown in Figure 5), it suggests a trend wherein girls trust humorous robots less, compared to boys.

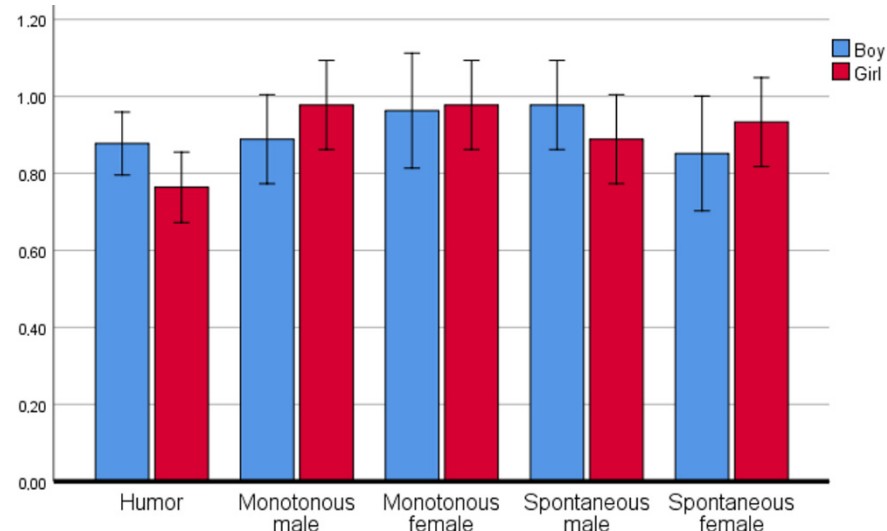

**Figure 5.** Mean trust levels per type of robot for boys and girls.

Previous experience with robots overall did not have a significant *(p < 0.05)* effect on children's trust levels per type of robot. However, children with experience showed a trend *(p ≤ 0.1)* wherein the robot was considered more trustworthy after the interaction compared to children who had no experience with robots, illustrated in Figure 6. This was only discovered for the humor and monotonous female characteristics.

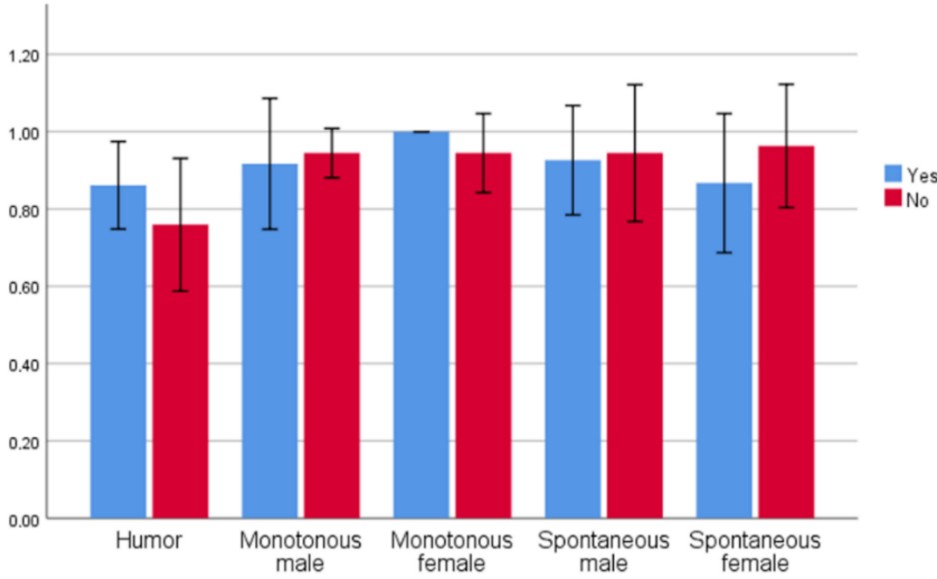

**Figure 6.** Mean trust levels per type of robot compared to robot experience (Yes = with experience; No = no experience).

*5.2. The Stress Experiment*

To explore the effect of a social robot on children's stress, we compared children's comfort levels before and after a child–robot interaction and compared it to a classical music intervention. We first conducted a General Linear Model (GLM) to explore the differences in children's comfort levels before and after the music and robot intervention, also considering age and gender. We found no significant differences between the robot

and the music intervention. However, just looking at the means of both groups, overall, children felt more comfortable after both interventions, illustrated in Figure 7.

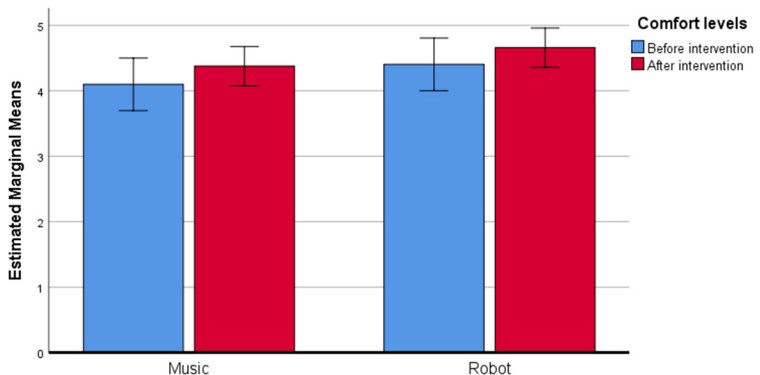

**Figure 7.** Mean comfort scores before and after the music and robot intervention.

During both interventions (music and robot) a research assistant made observations based on an observation scheme [51]. Figure 8 shows the frequencies of both interventions. Overall children in the robot groups were observed to smile more, compared to the music group. However, the discomfort scores and at ease scores were similar. Only three children frowned, which were all in the music group.

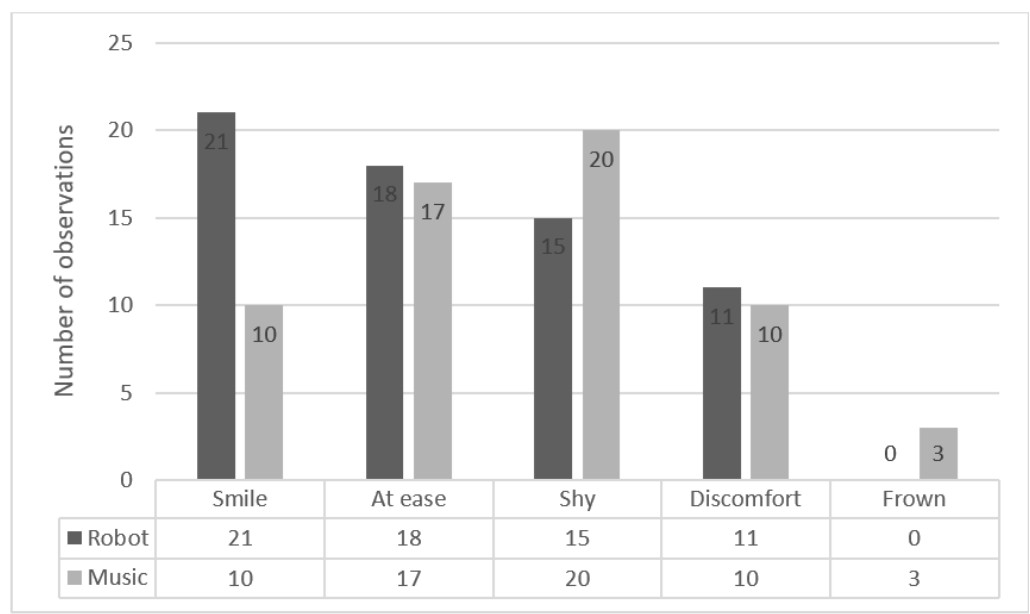

**Figure 8.** Overview of observation results.

We conducted a GLM test to explore the differences in observations for gender, age, and intervention type. We found some significant ($p = <0.05$) results for gender and discomfort ($p = 0.017$) wherein more girls ($n = 15$) were observed to show signs of discomfort compared to boys ($n = 6$). Significant results were also found for the type of intervention (robot or music) and smile ($p = 0.004$), wherein more children smiled in the robot's group ($n = 21$) compared to the music group ($n = 10$). Type of interaction and frown and age and frown showed trends ($p = <0.1$), however due to the small number of participants that frowned we have not discussed them in detail in this section.

Next to the observations in both groups, we also examined children's experiences with the robot. Five questions were asked to the children participating in the robot groups: if they considered the robot friendly; if they felt comfortable with the robot; if they would like

to be friends with the robot; if they liked the physical appearance of the robot; lastly whether they thought the robot would like to talk to them. To explore how children experienced the robot in relation to the changes in stress levels before and after the robot, we conducted a UNIANOVA test, taking into account gender and age. The experience of children had no significant effect on the changes in children's comfort levels. Children tended to rate the robot very positively in all aspects, see Table 1 below.

**Table 1.** Overview of the questionnaire results of the perspectives of children.

| Question | Answer | *n* | % |
|---|---|---|---|
| The robot is friendly. | Yes | 29 | 97% |
| | No | 1 | 3% |
| I felt comfortable with the robot. | Yes | 20 | 67% |
| | No | 10 | 33% |
| I would like to be friends with this robot. | Yes | 24 | 80% |
| | No | 6 | 20% |
| I like the robot's physical appearance. | Yes | 25 | 83% |
| | No | 5 | 17% |
| The robot wanted to talk with me. | Yes | 26 | 87% |
| | No | 4 | 13% |

## 6. Discussion

Earlier research has reported that social robots hold potential for supporting children's well-being during primary school. In this study, we examined the effect of multiple robot features on children's trust in social robots, and we explored the robot's ability to reduce children's stress levels. With this aim, we set up two experiments using the SAMbuddy robot cuddle. In this section, we first discuss the main results of our experiments in the light of existing research and provide our conclusions and recommendations for future research.

The main result for our experiment wherein we tested the impact of multiple robot features (intonation, male/female voice, and humor) on children's trust in a robot, was that, regardless of features, overall, children indicated high trust levels. This is consistent with the idea that social trust (as compared to competence trust) might be less affected by robot features [32], suggesting that robots designed to provide social and emotional support to children, such as SAMbuddy, do not benefit significantly from more human-like design features. Humor was the only feature that seemed to have a relatively negative effect on trust, especially for girls, and for children without prior experience with robots, although, even for these groups the mean trust scores can still be considered high. The relative negative effect of humor can be considered somewhat surprising since we expected humor to have a positive effect on trust, as reported in earlier research [34]. In addition, in HRI literature, humor seems to be mainly associated with positive outcomes such as users' perception of task enjoyment and robot personality (e.g., Niculescu et al., 2013). However, it should be noted that the positive impact of humor reported by [34] was based on human–human interaction and not human–robot interaction. Furthermore, in the humor scenario, only a female voice could be tested due to limited resources. We, therefore, encourage future research to study the effects on robots with male voices as we expect this could affect the level of trust differently compared to humorous female voices, see also the work of [58]. Furthermore, the sample size of the trust experiment (n = 55) can be considered small for such a multi-variable study. However, given the exploratory nature of our study, we consider our insights relevant for other researchers and robotic designers in the understanding of child–robot interaction and to create trustworthy child–robot interaction.

Some additional findings were discovered during the analyses of the results. The gender of the robot's voice did not influence the robots' trustworthiness. In the literature [22] gender has been shown to affect perceived trustworthiness in human–robot interaction. Furthermore, the intonation had no significant effect on trustworthiness. Earlier research has stated that intonation does play a part in gaining one's trust [35], however, this is not supported by our results.

The main result of our experiment wherein we tested the robots' ability to reduce children's stress compared to a classical music therapy session was that the robot was just as able to reduce stress as a classical music therapy session, although for both scenarios no significant reduction in stress levels was registered. Overall, children smiled more during the robot interaction, compared to the music session, although a considerable number of children did show some sign of discomfort while interacting with the robot. Girls were observed to show significantly more signs of discomfort compared to boys. Notably, children's comfort ratings were comparatively lower than all other scores, which has the surprising implication that, even though they feel somewhat uneasy around the robot, they still had a positive perception of it, and would be willing to trust it. This inconsistency could be attributed to different factors, that are known to affect subjective comfort judgements. For instance, novel objects and situations can be perceived as a potential stressor [59–61], and this is strongly modulated by individual differences in temperament and trait anxiety [60–62]. Similarly, perceived comfort has been shown to be affected by individual personality traits and gender [37,60,62], the effect of the latter being replicated in our study. From this perspective, these results highlight the importance of accounting for individual factors that affect an individual's subjective sense of comfort, and thus the extent to which they are willing to trust and collaborate with social robots.

Lastly, we found that almost all of the children (>95%) enjoyed using the social robot. This is in line with earlier research, and the novelty effect [63] is likely to be of influence here. Therefore, we encourage other researchers to study child–robot interaction over a longer period using a longitudinal approach so that the novelty effect can be mitigated as much as possible. One should also take into account the selection of participants in this study, which affected the generalizability of the results. The sample consisted of only Dutch children, which limited the cultural diversity and made it difficult to generalize towards children with a cultural background that differs from the Dutch culture.

Finally, creating trustworthy relationships between children and robots should not be done without considering moral implications. Having children share their social issues with robots can potentially assist in finding the necessary help they need to overcome the struggles that they are experiencing. However, as a result of this trustworthy relationship children might start to imagine that the robot really cares about them, and eventually be left feeling deceived [64]. Privacy issues also arise, such as what should the robot do with the secrets told by a child, and how should a robot be able to distinguish secrets from learning-related data. In this light, it is important to keep considering the ethical implications of social robots for educational purposes.

## 7. Conclusions

This study aimed to explore whether, humor, intonation, and gender play a role in gaining children's trust in social robots, and if social robots can reduce children's stress levels in primary education. Our results have shown the potential for robots to establish trustworthy interactions with children. Furthermore, we showed that children feel comfortable interacting with a robot designed for comfort and that social robots can be just as effective in reducing children's stress as a traditional classical music session. From a more practical perspective, it is noteworthy that both the robot and the classical music intervention have a comparable stress-reducing effect. This might suggest that the beneficial effects of social robots can be elicited to some extent by robots that are significantly less advanced and complex (and thus, less expensive) than other social robots available (e.g., PARO, AIBO, Pepper). This finding would suggest that simple, more affordable robots could be seen as

a viable option for stress reduction in situations where more advanced and sophisticated devices are not available. In the light of these results, we believe that social robots provide a new tool for children to open up their feelings, thereby enabling children to share negative experiences (such as being bullied) which would otherwise stay unnoticed.

**Author Contributions:** Conceptualization, M.H.J.S., S.L. and K.S.; methodology, M.H.J.S., S.L. and K.S.; validation, M.H.J.S., S.L., K.S. and D.F.P.V.; formal analysis, M.H.J.S. and D.F.P.V.; investigation, Y.O., J.O., T.U., M.S., T.v.d.K. and L.T.; resources, M.H.J.S., S.L., K.S. and M.H.J.S.; data curation, M.H.J.S., S.L. and K.S.; writing—original draft preparation, M.H.J.S., S.L. and K.S.; writing—review and editing, M.H.J.S., S.L., K.S. and D.F.P.V.; visualization, M.H.J.S., S.L., K.S. and D.F.P.V.; project administration, M.H.J.S., S.L. and K.S. All authors have read and agreed to the published version of the manuscript.

**Funding:** This research received no external funding.

**Institutional Review Board Statement:** The study was conducted according to the guidelines of the Declaration of Helsinki and in line with the institutional guidelines of the Institute of ICT, HU University of Applied Sciences Utrecht.

**Informed Consent Statement:** Informed consent was obtained from all subjects involved in the study.

**Data Availability Statement:** Data can be made available by the corresponding author or accessed via https://osf.io/h4jme/?view_only=1c27a4a25ff34dabaef5b9c3d515032e (accessed on 10 March 2022).

**Acknowledgments:** We are extremely grateful to all participants in our study. We also would like to thank the SAMbuddy team for lending their robot for testing, with special thanks to Nick van Breda, Raoul Postel, and Joanne Kroes.

**Conflicts of Interest:** The authors declare no conflict of interest.

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
