# Peer review of "A Trustworthy Robot Buddy for Primary School Children"

_mti, doi:10.3390/mti6040029_

Round 1

Reviewer 1 Report

The article has an interesting topic, it is structured logically and understandably, it contains all the elements of a scientific article.

The research questions are mentioned in the introduction and the authors direct the content of the article to answer them.In the introduction, the authors in the related work often mention the use of robots in the identification of bullying, but this element is later lost in the content.

Methods, materials and procedures are described clearly and comprehensibly, the number of attendees provides a relevant sample in the research to get relevant conclusions.

Based on the content of the article, I got the impression that the authors assumed that children would respond better to humour. However, it is difficult with humour, children at this age have very gentle and sensitive nuances. Every child considers something different to humour, and if the behaviour of the robot is not harmonized in its perception, it will cause a neutral or disturbing effect. In the future, authors should consider the appropriateness of this form of communication and find broader support in the literature.

The weakness of the second part of the research, which the authors are aware of, is the risk of novelty, which can affect the results. In the current implementation of the research, it is not possible to say whether the children enjoyed using the social robot or the novelty of using it. But it's nice that the authors are aware of this fact, mention it in the discussion and recommend to realize long-term study.

Within the description of the research, it would be appropriate to add the used questions of questionnaire (s) into the article, even though I tried to find them in the mentioned source, I did not succeed. Adding them to the article would improve the flow of reading.

Overall, I consider the article to be interesting and, given the demanding implementation of research in this area, also unique. Although the results of the research are not groundbreaking, I have not identified any serious errors in it and it can be a suitable material for researchers focused on this area.

Author Response

Dear reviewer,

Thanks for reviewing our paper, please see the attachment for the feedback breakdown and the responses.

Kind regards,

The authors

Reviewer 2 Report

The authors present findings about the trust perceived from a social robot, as well as the stress reduction from interacting with it, by children. Some of their results show deviation from common conceptions of social robotics, such as the robot's voice gender or intonation not having as much of an impact as before believed, and that simpler robots are as viable for these tasks as their costlier counterparts.

The manuscript is clearly written, with some minor adjustments needed:

- The test was carried out with Dutch children, which may suggest that these findings may be only applicable to the youth that lives in Netherlands, and cannot be generalized to other cultures. This reviewer suggest adding text to clarify this to the reader.

- In lines 306-307 the authors state that: "children with experience showed a trend (p=<0.1) wherein the robot was considered more trustworthy after the interaction compared to children who had no experience with robots, illustrated in Figure 6." However, Figure 6 shows shows that in average only 2 cases out of 5, children with experience trusted more than those without experience. The authors need to provide insight of how this is the case, while there is a trend showing the opposite.

- There is a considerable dip in the positive scores of the comfort question in Table 1 (67%) compared to the relatively high scores in the other questions (between 80 and 97%). Is this consistent with the "trust" experiment findings where there children showed relatively high levels of trust? The authors should clarify what is the difference between "trust" and "comfort" in their experiment.

- Although the authors do acknowledge the reason why a "humorous male" voice was not tested (limited resources), it would be of benefit to the reader that the authors provide insight of what they suspect would happen.

- Although the manuscript has virtually no grammar/style issues, there are some things the authors should be consistent in:
. The use of a "," instead "." in the formatting of decimal numbers.
. The styling of the "p" variable, when sometimes is in italics, and others not.

Author Response

(The authors gave the same response as above.)
